# Hand choice is unaffected by high frequency continuous theta burst transcranial magnetic stimulation to the posterior parietal cortex

**Aoife M. Fitzpatrick**[1,2], **Neil M. Dundon**[3], **Kenneth F. Valyear**[2,4] *

1 Neuroscience and Behaviour Laboratory, Italian Institute of Technology, Rome, Italy, 2 School of Human and Behavioural Sciences, Bangor University, Bangor, United Kingdom, 3 Brain Imaging Center, Department of Psychological and Brain Sciences, University of California, Santa Barbara, CA, United States of America, 4 Bangor Imaging Unit, Bangor University, Bangor, United Kingdom

* k.valyear@bangor.ac.uk

**Data Availability Statement:** Online data repository: Link: https://osf.io/xkmgq/ DOI: 10.17605/OSF.IO/XKMGQ.

## Abstract

The current study used a high frequency TMS protocol known as continuous theta burst stimulation (cTBS) to test a model of hand choice that relies on competing interactions between the hemispheres of the posterior parietal cortex. Based on the assumption that cTBS reduces cortical excitability, the model predicts a significant decrease in the likelihood of selecting the hand contralateral to stimulation. An established behavioural paradigm was used to estimate hand choice in each individual, and these measures were compared across three stimulation conditions: cTBS to the left posterior parietal cortex, cTBS to the right posterior parietal cortex, or sham cTBS. Our results provide no supporting evidence for the interhemispheric competition model. We find no effects of cTBS on hand choice, independent of whether the left or right posterior parietal cortex was stimulated. Our results are nonetheless of value as a point of comparison against prior brain stimulation findings that, in contrast, provide evidence for a causal role for the posterior parietal cortex in hand choice.

## 1. Introduction

Decision-making for the purpose of acting involves deciding both which actions to perform–action selection–and how to perform them. Sensorimotor-based models of action selection suggest that the same brain mechanisms responsible for the parameterisation of possible actions in sensorimotor terms also mediate action selection [1–3]. Co-opting the same neural 'currency' to both parameterise and select actions is not only efficient, but enables estimates of predicted sensorimotor costs (for e.g., energetic demands related to biomechanical factors) to directly inform choices [4]. Evidence supporting sensorimotor-based models of action selection spans multiple domains, including animal neurophysiology [5–7], cortical inactivation [8, 9], and microstimulation [10, 11], human behaviour [12–14], and neuroimaging [15, 16].

Competition between neural populations that code for different actions is a common feature of sensorimotor-based models of action selection. For example, both the Affordance Competition Hypothesis [17, 18] and the Dynamic Neural Fields model of Christopoulos et al.

**Funding:** The author(s) received no specific funding for this work.

**Competing interests:** The authors have declared that no competing interests exist.

[2] suggest that action choices are made by resolving competition between simultaneously activated neural populations that define the spatiotemporal parameters of possible actions. Each of the two models specify the following details. These neural populations are located within reciprocally connected areas of posterior parietal and frontal premotor cortices. Populations encoding similar parameters are mutually excitatory while those encoding distinct parameters are mutually inhibitory. Once the activity of one population exceeds a particular (threshold) level the spatiotemporal parameters of the actions encoded by this population are selected. Inputs from other brain areas enable various other (i.e. non-sensorimotor) decision variables (e.g. arbitrary learned associations) to weigh-in, directly influencing the activity of competing populations and thus influencing selection.

Experimental methods that enable focal disruption of brain function provide powerful ways to test sensorimotor-based models of action selection. Reversible inactivation of monkey parietal reach region (PRR) dedicated to the planning and control of arm movements selectively impairs arm- but not eye-movement choices [9], while inactivation of the nearby lateral intraparietal area (LIP) important for the planning and control of eye movements predominantly affects eye- over arm-movement choices [8]. These data provide strong support for sensorimotor-based models of action selection. Action choices involving different effectors rely on brain areas necessary for the planning and control of those effectors.

Oliveira et al. [19] used an analogous 'knock-out' approach to test whether posterior parietal areas important for arm control in humans are also important for deciding which hand to use to perform a reaching task. Participants used either hand ("free choice") to reach to visual targets in different parts of space. Using single-pulse TMS, reach-selective areas in posterior parietal cortex (PPC) were targeted during the premovement planning phase (100ms after target presentation). TMS to the left hemisphere PPC increased the likelihood of reaches made with the left hand. TMS to the right hemisphere PPC, conversely, did not reliably influence hand choice. The reason for this asymmetry remains unclear.

Motivated by the evidence introduced above, we developed a sensorimotor-based model of hand selection: The Posterior Parietal Interhemispheric Competition (PPIC) model (Fig 1A). The PPIC model makes two assumptions: (1) there are populations of neurons within the posterior intraparietal and superior parietal cortex (pIP-SPC) of both the left and right hemispheres that specify sensorimotor parameters for actions in hand-specific coordinates, and (2) within each hemisphere, more of these neural populations code for actions with the contralateral hand.

Otherwise, the mechanics of the model are taken directly from the Affordance Competition Hypothesis [17]. Populations of neurons encoding similar actions with the same hand excite one another while those encoding actions with the opposite hand (and those encoding dissimilar actions) inhibit one another, and the strength of influence of a given neural population scales nonlinearly with its current level of activity. Hand (and action) selection is determined when the activity of one of these neural populations reaches suprathreshold levels.

We recently tested the PPIC model using functional MRI and found supporting evidence [16]. Areas within bilateral pIP-SPC were significantly more active during reaching actions involving free choice of which hand to use compared to when hand use was pre-instructed. This pattern of choice-selectivity was found bilaterally in pIP-SPC for actions made with either hand; although, within each hemisphere, actions with the contralateral hand evoked the strongest responses. Consistent with a competitive process, fMRI activity levels were elevated for responses to targets that represented more ambiguous choices, paralleled by increased response times, and these effects were specific to areas within the PPC. The findings are consistent with the PPIC model and the hypothesis that the PPC plays an important role in deciding which hand to use to perform actions.

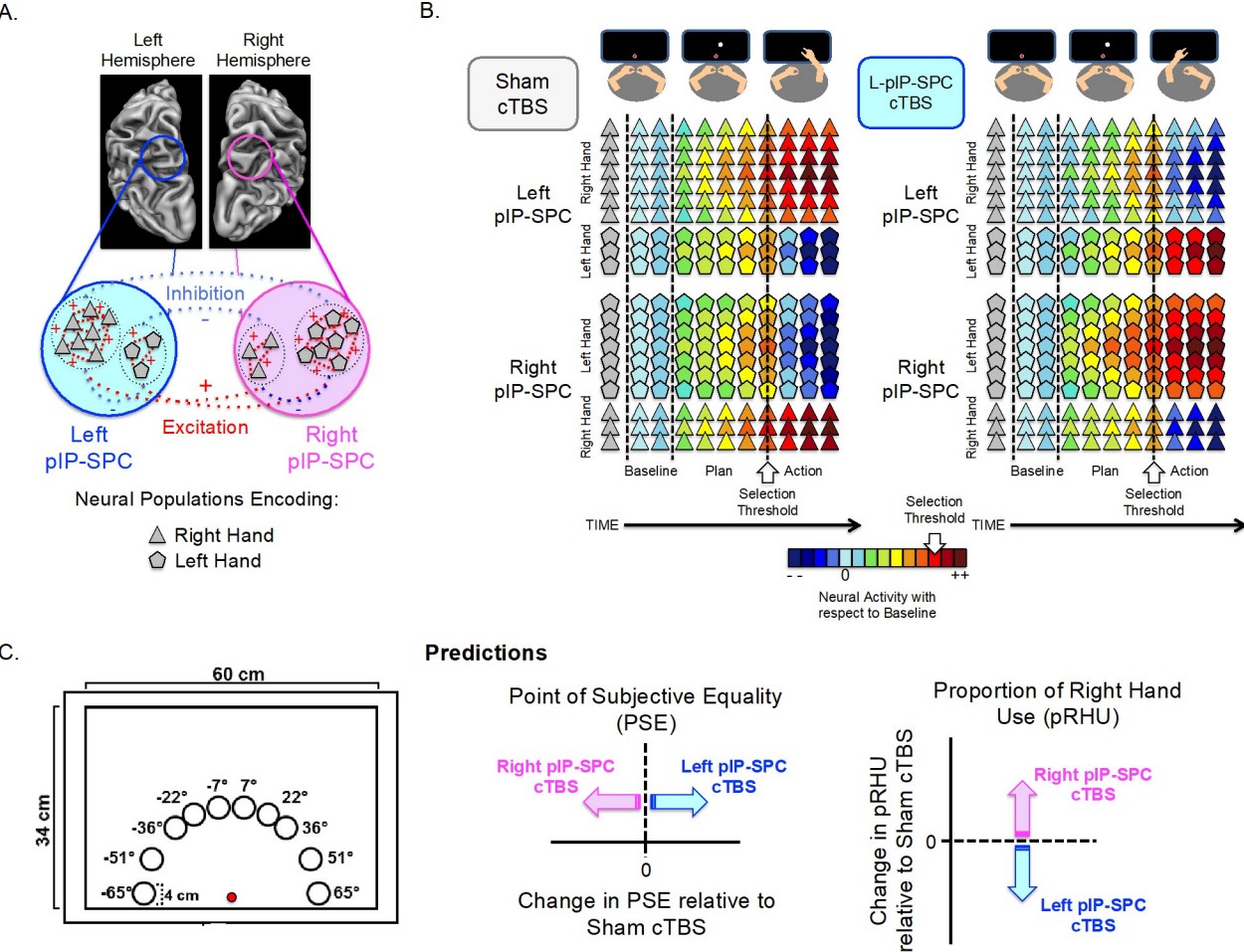

**Fig 1. The PPIC Model, methods and predictions. (A)** *The PPIC model.* Populations of neurons in pIP-SPC encode actions in hand-specific terms. Within each hemisphere, the contralateral hand is overrepresented. Neural populations encoding actions with the same hand excite one another while those that encode actions with the opposite hand inhibit one another. **(B)** An example of the predicted changes for a single trial involving a target presented at 7˚ following Sham (left panel) and L-pIP-SPC (right panel) cTBS. After the target is presented, during premovement planning, the activity of all cell-types increases. In the Sham cTBS condition, neural populations encoding the right hand show a steeper rate of increase, and reach suprathreshold activity first. The right hand is selected to perform the task. CTBS to left pIP-SPC reduces both its excitatory potential and (largely inhibitory) influence on the right pIP-SPC. Consequently, cells dedicated to the left hand in the right pIP-SPC now reach suprathreshold activity first, and thus the left hand is selected to perform the task. **(C)** *Experimental set-up.* Schematic representation of targets for reaching, arranged symmetrically around the midline of the display. The red circle represents fixation. *Predictions.* The PPIC model predicts that cTBS to left/right hemisphere pIP-SPC will reduce the likelihood of choosing the contralateral hand. Compared to Sham stimulation, a rightward (positive) shift in the PSE and a decrease in the proportion of right-hand use is predicted following cTBS to the left hemisphere pIP-SPC (shown in blue). The opposite pattern is predicted for cTBS to the right hemisphere pIP-SPC (shown in pink).

In the current study we use a high-frequency repetitive TMS protocol known as continuous theta burst stimulation (cTBS) to evaluate the PPIC model. When applied to primary motor cortex, cTBS has been shown to reduce cortical excitability for up to 60 minutes [20]. These suppressive effects are thought to reflect reduced synaptic efficacy [21]. We apply cTBS to test the PPIC model: cTBS to unilateral pIP-SPC should decrease the probability of selecting the hand contralateral to stimulation.

Two features of the PPIC model motivate this hypothesis (Fig 1). First, a greater proportion of cells in pIP-SPC encode the selection and use of the contra- vs. ipsilateral hand. The dampening effects of cTBS on synaptic efficacy are expected to disproportionately impact these cells, driving down their excitatory potential, and as a consequence decrease the likelihood of

selecting the hand contralateral to the site of stimulation. Second, decreased excitatory potential in these cells will decrease their activity-dependent inhibitory drive on those (opponent) cells encoding the ipsilateral hand, found predominately in the opposite hemisphere. This would also be expected to reduce the likelihood of using the hand contralateral to the site of stimulation. We pre-registered these predictions at aspredicted.org (https://aspredicted.org/ IRX_BZV) before data collection for the study began.

We used the behavioural paradigm developed by Oliveira et al. [19]. This paradigm enables highly sensitive subject-specific measures of hand choice. For each participant the point in target space where the choice of either hand is equally probable–the point of subjective equality (PSE)–can be estimated. Consistent with a competitive process underlying hand choice, Oliveira et al. [19] demonstrate that the PSE defines an area of space with greater choice costs. They found that response times to initiate reaches to targets near the PSE were larger than those to more lateralised targets, and, critically, these differences were specific to when hand choice was required; when hand choice was predetermined these differences were not observed. TMS induced the largest change in hand choice for responses to targets near the PSE.

The PPIC model and our predictions for the current study are only partly consistent with the results of Oliveira et al. [19]. Specifically, as noted above, the evidence from Oliveira and colleagues [19] suggests that the involvement of PPC in hand choice is left-lateralised; stimulation of the left but not the right PPC influenced hand choice. Accordingly, based on these results, we may expect to find that cTBS to the left but not the right PPC will lead to a change in hand choice behaviour in our reaching task.

Following completion of the current study, new evidence to suggest a causal role for the PPC in hand choice was provided by Hirayama et al. [22]. Using a similar behavioural protocol, also modelled after Oliveira et al. [19], Hirayama et al. [22] found that bilateral transcranial direct current stimulation (tDCS) over the left and right hemisphere PPC leads to stimulation aftereffects that increase the likelihood of using the left hand, expressed as a rightward shift in the PSE. We provide detailed consideration of this study and their findings within our Discussion, together with our current results.

## 2. Material and methods

### 2.1 Pre-registration

The study was pre-registered via aspredicted.org (https://aspredicted.org/IRX_BZV). Pre-registration included our principal predictions and analyses plan.

Power analysis (using G*Power; [23]) was used to estimate sample size on the basis of the effect size (d = 0.76) calculated from Oliveira et al. [19], discussed above. The results suggest that a sample size of 20 participants is sufficient to detect an effect-size of d = 0.76, with 95% power using a paired-samples t-test with alpha at 0.05.

### 2.2 Participants

Twenty-six individuals (14 males, 12 females, M = 22.54 years ± 3.24 SD) participated in the experiment. Handedness was qualified using a modified Waterloo Handedness Inventory [24]. Here, we report the data from 20 right-handed participants (10 males, 10 females; mean age = 22.90 years ± 3.48 SD; Waterloo Handedness scores: range = 16–30, median = 25); with all participants included, the main statistical outcomes are the same (see S1 Table in S1 File and S2 Table in S2 File Supplementary statistical analyses). Excluded participants include three left-handed participants (Waterloo Handedness scores: -30, -27, -11), two self-reported strategy-users, and one individual who experienced moderate-adverse-effects of TMS.

Participants completed a single MRI session involving an anatomical scan followed by three sessions of TMS and behavioural testing. Each TMS-behavioural test took approximately one hour and twenty minutes to complete, separated by a minimum of 1 week (M = 7.60 days; SD = 2.26). All participants provided informed consent in accordance with the Bangor University School of Psychology Ethics Board, and were naïve to the goals and predictions of the study. All participants had normal or corrected-to-normal vision, with no MRI/TMS contraindications. Participants were financially compensated.

## 2.3 Experimental setup and materials

Participants were seated ~50cm from a 65cm x 45.5cm vertical touchscreen monitor (1920 x 1080 resolution), centred with respect to their mid-sagittal plane. At the start of a trial, the left and right index fingers held down two start keys (2.2cm x 3.3cm), fixed to a table 30cm from the monitor, aligned with the centre of the monitor. Targets were 4cm-diameter white circles presented against a uniform black background. Targets were presented at 10 positions relative to midline: -65, -51, -36, -22, -7, 7, 22, 36, 51, and 65 degrees, jittered by a 2D Gaussian kernel (SD = 0.5cm) (Fig 1C). A similar target configuration was used previously [19, 22, 25]. Targets were equidistant (30cm) from the centre of start-keys, and comfortably reachable with either hand. A central fixation point (0.2cm x 0.2cm) was displayed at 5cm from the base of the monitor screen. The experiment was controlled in Matlab (r2015b) using the Psychophysics Toolbox extensions [26–28].

## 2.4 Procedure

**2.4.1 Transcranial magnetic stimulation.** TMS was delivered using a Magstim Rapid Plus stimulator with a 70mm figure-of-eight coil. Coil localization was performed using the BrainSight frameless stereotaxic neuronavigation system (BrainSight Software, Rogue Research Inc., Montreal, Quebec, Canada, version 2.3.10; Polaris System, Northern Digital Inc., Waterloo, Ontario, Canada) and individual participant MRI data.

T1-weighted anatomical MRI data were collected on a 3T Philips Achieva scanner using a multiplanar rapidly-acquired gradient echo (MP-RAGE) pulse sequence: time to repetition = 1500ms; time to echo = 3.45ms; flip angle = 8˚; matrix size = 224 by 224; field of view = 224mm; 175 contiguous transverse slices; slice thickness = 1mm; in-plane resolution = 1mm by 1mm.

High-frequency repetitive continuous theta burst stimulation (cTBS) was used to evaluate the PPIC model and the hypothesis that bilateral pIP-SPC is critically involved in hand choice. Following the protocol introduced by Huang et al. [20], cTBS involved the application of 'bursts' of three TMS pulses at 50Hz, with an inter-burst frequency of 5Hz for 40s (600 pulses).

First, active motor thresholds were defined per participant. The participant's anatomical MRI was used to estimate the location of the hand area in the primary motor cortex of the left hemisphere, identified as the characteristically inverted-omega-shaped 'hand knob' within the central sulcus [29]. With the coil held tangentially on the scalp surface, handle oriented posteriorly and angled laterally at approximately 45˚ from the midline, single TMS pulses were delivered while electromyographic recordings were measured from the contralateral first dorsal interosseous (FDI) muscle. Starting at this location, single pulses were delivered while monitoring the electromyographic activity from the FDI. The coil was then moved in small increments, until the location where a maximal amplitude motor evoked potential (MEP) from the FDI was defined–the motor hotspot.

With the coil positioned at the motor hotspot, active motor thresholds were defined as the minimum stimulator intensity wherein peak-to-peak MEP amplitudes of greater than 200μV

were elicited in 5/10 consecutive trials while the subject was voluntarily contracting their FDI muscle at 20% maximal force using visual feedback [30]. For subsequent cTBS, the intensity of the stimulator was set to 80% of the participant's active motor threshold.

To target the L- and R-pIP-SPC for cTBS we used a combination of functional and anatomical guidelines. First, we used results of our previous fMRI study identifying hand-choice-selective responses in L- and R-pIP-SPC [16]. Specifically, for L- and R-pIP-SPC stimulation the TMS coil was moved to the coordinates of the hotspots of activity identified by the group-level contrast of Choice > Instruct conditions in Fitzpatrick et al. [16]. Second, if necessary, the coil position was then adjusted so that the target trajectory passed through the medial bank of the intraparietal sulcus, within the superior parietal cortex. On the basis of our fMRI results, the R-pIP-SPC target was marginally more posterior and medial than the L-pIP-SPC target (see Discussion, Fig 5). In line with previous applications of TMS to the PPC [31–33], the coil-handle orientation was set posterior and approximately parallel with the midline.

Sham cTBS involved positioning the coil over either the L- or R-pIP-SPC using the same approach described above yet with coil surface angled 90˚ from the scalp during stimulation. With this approach, the feeling of the coil on the surface of the head and the sounds made from discharging the coil are similar to active cTBS, yet any stimulation that penetrates the scalp (via the wing of the coil) is presumed ineffective–i.e. unlikely to meaningfully influence cortical physiology [34, 35]. Coil position over the L- or R-pIP-SPC for sham stimulation was counterbalanced across participants.

The stimulation aftereffects of the cTBS protocol are reported to stabilise approximately five minutes following stimulation [20]. Participants began behavioural testing after a five-minute period.

**2.4.2 Behavioural testing.** Trials began with participants in the start position, holding down each of the start keys with their index fingers. Participants were instructed to fixate the central fixation point. When both start keys were depressed, a 400ms-duration tone was played to alert participants that the trial had started. This was followed by a variable delay (200/400/600/800ms, randomly ordered). Next, a target appeared at one of the 10 positions in the target array. Participants were instructed to reach to contact the target with the index finger of one hand, as quickly and accurately as possible. They were told that that they could move their eyes freely during reaching. No explicit instruction regarding the possible correction of their reach trajectories was given. Target onset was coincident with the removal of the fixation point. Targets were removed after movement onset, triggered by the release of a start key. The next trial began as soon as the participant returned to the start keys.

Two additional types of trials were included: two-target and fixation-catch conditions (following [19, 22, 25]). In the two-target condition, two targets were presented at the most peripheral edges of the target array (i.e. at -65/65 degrees ± jitter). Participants were instructed to use both hands to contact targets, and to move each hand together at the same time. The fixation-catch condition involved the presentation of a single target near fixation; again, participants were instructed to use both hands to contact this target, and to move each hand together. These conditions were included to minimize the likelihood that participants would persist in the use of only one hand during single-target conditions. Additionally, the fixation-catch condition was expected to strengthen the likelihood that fixation would be maintained at the start of each trial, prior to target onset.

Participants completed six blocks of 145 trials per session. Two blocks were completed pre-cTBS, and four blocks were completed post-cTBS. A custom Matlab (R2011b) script was used to create trial sequences wherein trial (t) history (t– 1) was balanced according to (1) condition and (2) target position for single-target conditions. Thus, each experimental block comprised 120 single-target trials, 12 per target position, and 24 two-target and fixation-catch trials,

balanced for condition history. A unique trial sequence was generated per block. The first trial of each block was an additional, randomly selected trial, not controlled for history. Data from the first trial of each block, two-target and fixation-catch conditions were excluded from analyses. Unless specified, pre-stimulation data were excluded from analyses (i.e. Sham cTBS was used as the control).

After the final cTBS-behavioural session participants completed (1) the Waterloo Handedness Inventory and (2) were asked if they "used a specific strategy, or rule" to decide which hand to use during behavioural testing.

## 2.5 Dependent measures and analyses

Study pre-registration included outlier removal procedures: Outliers were defined as ± 2.5 standard deviations from the group mean, per statistical test, and removed from further analyses. Results from non-outlier-removed analyses are reported in the Supplemental Materials (S1 Table in S1 File and S2 Table in S2 File). All results were considered significant at p < 0.05.

**2.5.1 Hand choice.**   Hand choice was measured using button-release data, and, if unclear (e.g. for trials involving multiple button releases), confirmed using video data. Consistent with previous investigations [19, 25], hand choice was tested using three analyses methods.

First, for each participant, a psychometric function [36] was computed according to their hand choice behaviour (on single-target conditions) per target location, and the theoretical point in space where the participant was equally likely to use either hand–the PSE–was determined. Specifically, PSEs were estimated by fitting a general linear model to each participant's hand choice data. The model included target positions and a constant term, and used a logit-link function to estimate the binomial distribution of hand choice responses (1 = right | 0 = left). Model coefficients were evaluated at 1,000 linearly spaced points between the outermost values of the target array (i.e. ± 65 degrees). The value closest to a 0.50 probability estimate was defined as the PSE. The model was fitted separately per individual, per cTBS condition. Resultant PSEs per cTBS condition were then evaluated using a repeated-measures ANOVA (rmANOVA).

Two additional analyses were performed. Hand-choice data expressed as proportions of right-hand use were arcsine transformed, calculated as the arcsine square root of the proportions. The arcsine transformation stretches the upper and lower ends of the data. This makes the distributions more symmetrical and reduces problems with violations of the assumption of normality. The transformed proportions were then tested using two separate rmANOVAs, with cTBS condition as the single fixed factor.

The first of these models was used to test for cTBS effects on the mean arcsine transformed proportions of right-hand-use collapsed across all target locations. The second model also tested the mean arcsine transformed proportions of right-hand-use yet restricted to those targets that bound the PSE, as defined per individual on the basis of Sham-cTBS. This final approach is consistent with that used by Oliveira et al. [19], and, as discussed above, was expected to comprise the most sensitive test of cTBS effects on hand choice.

**2.5.2 Response time.**   Response time (RT) was measured using button-release data, and defined as the time taken to initiate a reach after target onset (in milliseconds, ms). We did not pre-register analyses and predictions regarding cTBS effects on response times. The push-pull characteristics of the PPIC model, however, make the following predictions.

By reducing the excitatory potential of neural populations underlying the site of stimulation, cTBS to unilateral pIP-SPC is predicted to slow the rates of excitation to reach selection thresholds for the contralateral hand (see Fig 1B). The contralateral hand is disproportionately impacted since neural populations representing the contralateral hand are overrepresented. As

a consequence, prolonged RTs to initiate reaches with the contralateral hand are predicted. Further, as a consequence of the rivalry between hemispheres, these effects are also predicted to result in faster RTs to initiate reaches with the limb ipsilateral to stimulation. Reduced excitatory potential of neural populations in one hemisphere will also reduce their inhibitory drive on those neural populations in the other hemisphere.

Response times per hand per cTBS condition were evaluated using a rmANOVA. Given that bimanual two-target catch trials always occurred at the most extreme lateral positions of the target array (± 65 degrees), single-target responses to these targets were excluded from RT analyses.

## 3. Results

Data reported include right-handers without strategy use (N = 20). Results from the complete dataset, including left-handers (N = 3), right-handers who reported strategy use (N = 2), and non-outlier-removed analyses are provided as Supplementary Materials (S1 Table in S1 File and S2 Table in S2 File). Participants made few errors, affecting <1% of the total number of trials (see S3 File Participant errors).

### 3.1 Hand choice

Fig 2A provides a descriptive overview of the hand choice results as a function of target location and cTBS condition. The data are expressed as the mean proportions of right-hand use (RHU). The figure shows both the group median and interparticipant distribution per target location and cTBS condition. Participants typically use their left hand to contact targets on the left side of space and their right hand to contact targets on the right side of space; target -7˚ shows the most variation in hand choice behaviour. This overall pattern is consistent across cTBS conditions.

As a main strength of this behavioural paradigm highly sensitive subject-specific measures of hand choice can be quantified. The PSE defines a theoretical point in target space where the choice of either hand is equally likely. Individual-level data with resultant curve-fits used to estimate PSEs per condition are shown for two participants (Fig 2B). The PSE is expected to reflect an area of target space with high choice costs; response time data support this expectation, see Section 3.2.

Inconsistent with our predictions, we find no evidence for changes in hand choice following cTBS. Results of a rmANOVA of PSEs reveal no significant differences between cTBS conditions (F(2, 36) = 0.56, p = 0.58, $\eta^2_p$ = .03) (Fig 2C). The group mean PSEs are near target -7˚ for all conditions. Both L- and R-pIP-SPC stimulation conditions show a small (< 2˚) and inconsistent rightward (positive) shift in group-mean PSE estimates relative to Sham-cTBS.

Analyses of arcsine transformed proportions of RHU reveal similar results. First, considering responses to all targets, we find no significant differences between cTBS conditions (F(2, 36) = 0.71, p = 0.50, $\eta^2_p$ = 0.04) (Fig 3A). Second, restricting our analyses to those data that bound the PSEs, as defined per participant, we again find no reliable effects of cTBS (F(2, 38) = 1.09, p = 0.35, $\eta^2_p$ = 0.05) (Fig 3B). Both results suggest reduced right-hand choice following real cTBS, yet these effects are statistically unreliable.

We performed a final set of analyses using No-cTBS as the baseline measure of hand choice, rather than Sham-cTBS. Although logically motivated, these analyses were not pre-planned, and as such, we prefer to restrict our report of these analyses to supplementary materials, to be interpreted with due caution (see S4 File Additional analyses. No cTBS-baseline).

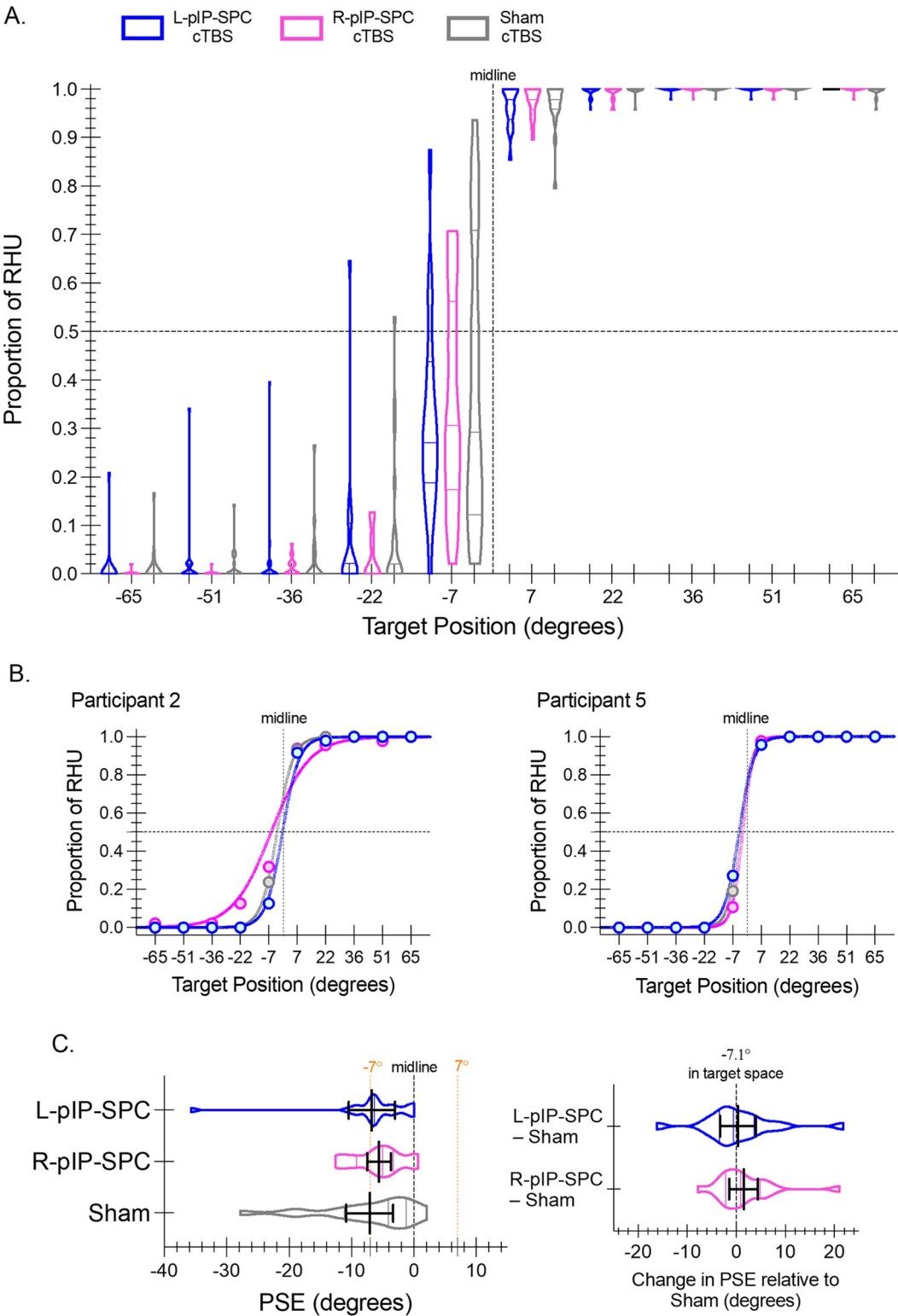

**Fig 2. Hand choice. (A)** Violin plots depict the interparticipant distribution of hand choice data across target locations expressed as the proportions of right-hand use (RHU) for L-pIP-SPC (blue), R-pIP-SPC (pink), and Sham (grey) cTBS conditions. Within each violin plot the median and upper and lower quartile values are indicated. A vertical dashed line depicts the midline of the display (0˚). A horizontal dashed line shows the point of equal proportion (0.50) of left- and right-hand use. **(B)** Individual-level data for two participants with resultant curve-fits per stimulation condition, used to estimate

PSE values, are shown. Filled circles show the proportion of RHU per target location per condition. **(C)** Violin plots show the interparticipant distribution of mean PSEs per condition. Solid black lines indicate group means with 95% confidence intervals. The locations of the midline and targets -7˚ and 7˚ are shown for reference. *Inset.* Difference scores of PSEs for each condition relative to Sham cTBS are shown as violin plots. Group means and 95% confidence intervals are overlaid.

### 3.2 Response times

Fig 4 illustrates response time results as a function of hand and cTBS condition (Fig 4A). We find no significant main effect of hand (F(1, 18) = 0.73, p = 0.40, $\eta^2_p$ = 0.04), cTBS condition (F(2, 36) = 0.90, p = 0.42, $\eta^2_p$ = 0.05), and no significant interaction (F(2, 36) = 0.41, p = 0.66, $\eta^2_p$ = 0.02). The interaction result is inconsistent with our predictions, where a relative increase in RTs for reaches made with the hand contralateral to the site of cTBS, accompanied by a relative decrease in RTs for reaches made with the ipsilateral hand, was expected.

RT data nonetheless support expectations regarding high choice costs around the PSE (Fig 4B). RTs to targets that bound the PSE (as defined by Sham-cTBS) are reliably greater than those to more lateral targets (at ± 51 degrees); we find a significant main effect of target location (F(1, 18) = 9.05, p < 0.01, $\eta^2_p$ = .34). The main effect of cTBS and the interaction between target location and cTBS are not significant (respectively: F(2, 36) = 0.72, p = 0.49, $\eta^2_p$ = 0.04; F(2, 36) = 0.67, p = 0.52, $\eta^2_p$ = 0.04).

These results provide important confirmatory evidence consistent with previous findings [16, 19, 22, 25]. Consistent with a competitive deliberation process, participants take more time to reach to targets that surround the PSE relative to targets at more lateral positions in left and right hemispace.

## 4. Discussion

In this study we apply a high-frequency TMS protocol known as cTBS to the left and right posterior parietal cortex (PPC), separately, and test its effects on hand choice. CTBS was expected to reduce excitability of the targeted area. According to the PPIC model, reduced excitability of the left PPC should decrease the probability of using the right hand (increase left hand use), whereas reduced excitability of the right PPC should decrease the probability of using the left

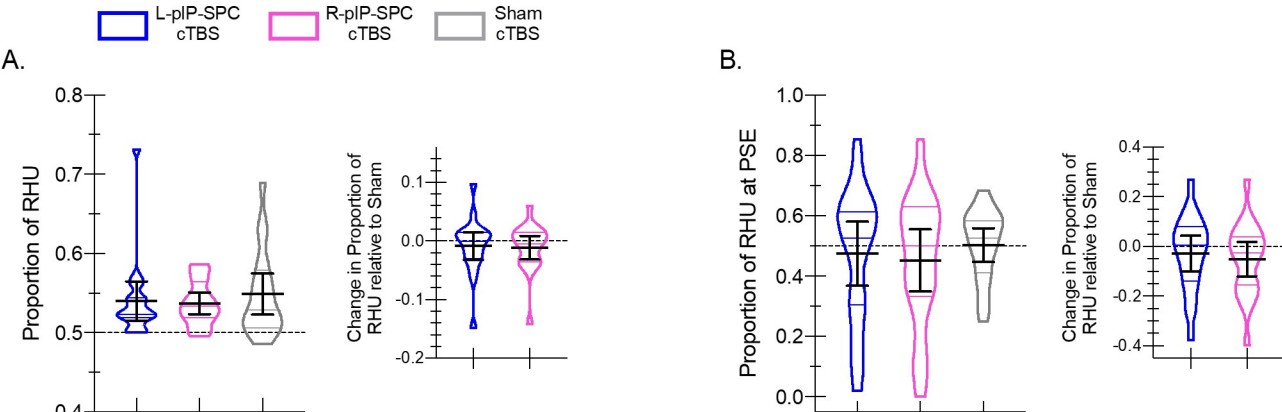

**Fig 3. Hand choice: Proportions of right-hand use.** **(A)** Violin plots show the interparticipant distribution of hand choice data collapsed across all targets expressed as the proportions of right-hand use (RHU) for L-pIP-SPC (blue), R-pIP-SPC (pink), and Sham (grey) cTBS conditions. Within each violin plot the median and upper and lower quartile values are indicated. Solid black lines indicate group means with 95% confidence intervals. *Inset.* Difference scores show the proportion of RHU per condition relative to Sham cTBS. Data are shown as violin plots with group means and 95% confidence intervals overlaid. **(B)** Same as in **(A)** yet restricted to those targets that bound the PSE.

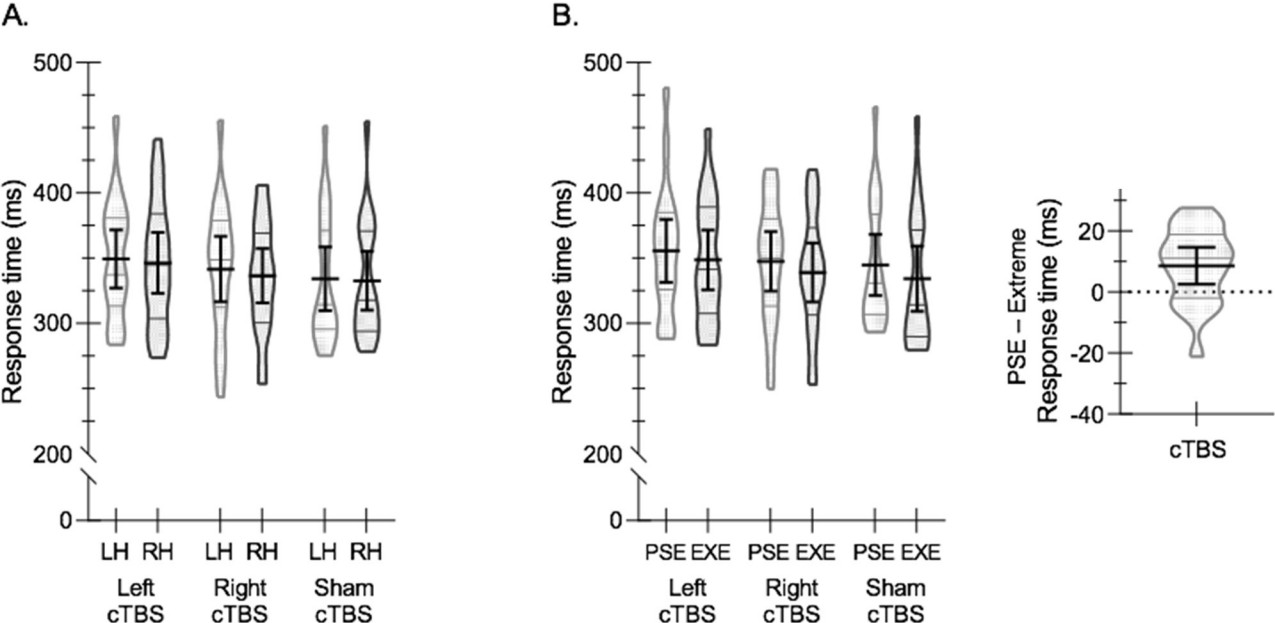

**Fig 4. Response times. (A)** Violin plots show the interparticipant distribution of mean response times (RT) for the left hand (LH) and right hand (RH) for L-pIP-SPC (Left cTBS), R-pIP-SPC (Right cTBS), and Sham (Sham cTBS) conditions. Within each violin plot the median and upper and lower quartiles are indicated. Solid black lines indicate group means with 95% confidence intervals. **(B)** Violin plots show the interparticipant distribution of mean RTs for targets bounding the PSE and at the lateral periphery (EXE) (±51 degrees). Greater RTs for reaching to PSE targets are evident across all cTBS conditions. *Inset*. The difference in RTs to targets that bound the PSE relative to those at the EXE, collapsed across cTBS conditions. The difference score is shown as a violin plot with the mean and 95% confidence intervals overlaid. The results indicate greater choice costs when reaching to targets near the PSE.

hand (increase right hand use) (Fig 1). Instead, we found no evidence that cTBS influenced hand choice. Participants made similar choices about which hand to use following real compared with Sham cTBS, independent of whether the left or right PPC was targeted. Response times to initiate actions were also unaffected by cTBS. The expected pattern of increased response times for targets near a given participant's point of subjective equality, where the choice of either hand was equally probable, was observed.

Our results are inconsistent with the PPIC model, and with the results of two previous brain stimulation studies which—using a similar behavioural protocol as we have used here, but different brain stimulation methods—demonstrate a causal role for the PPC in hand choice. Oliveira et al. [19] show that a single TMS pulse delivered to the left (but not right) PPC during the premovement phase, 100ms after the onset of a visual target for reaching, increases the likelihood of using the left hand, characterised by a rightward shift of the PSE in target space. More recently, and following the completion of our study, Hirayama et al. [22] show that bilateral tDCS to the PPC with cathodal stimulation over the left hemisphere and anodal stimulation over the right hemisphere leads to stimulation aftereffects that increase the likelihood of using the left hand, characterised by a rightward shift of the PSE in target space. We focus our discussion on the methodological differences between these two studies and our own, and how these differences may account for our inconsistent results.

### 4.1 Differences in brain stimulation methods

The discrepancy between the results of our study and the respective findings of Oliveira et al. [19] and Hirayama et al. [22] may be attributable to the use of different brain stimulation

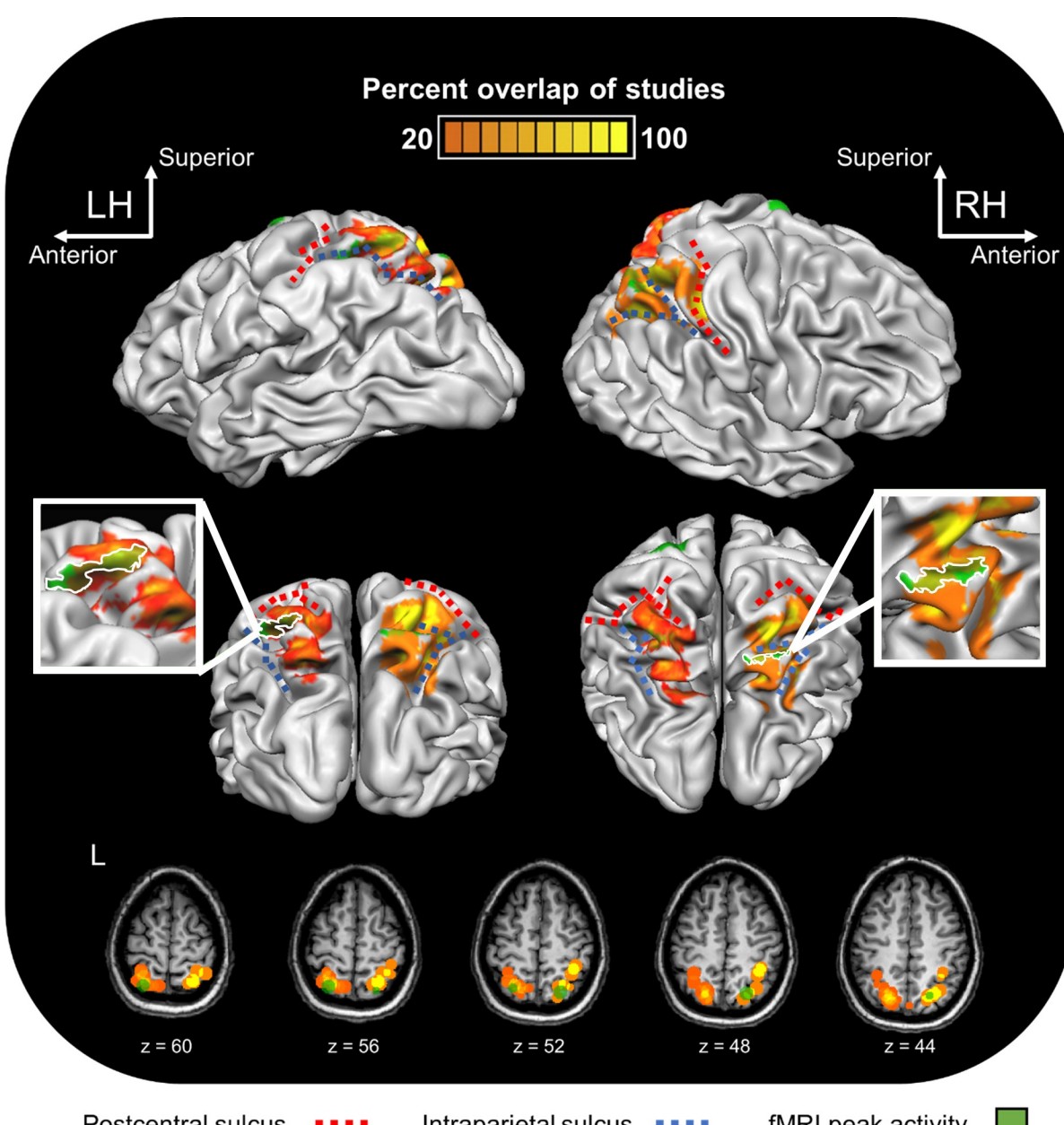

**Fig 5. Synthesis and visualisation of prior brain imaging and stimulation results involving reaching.** 3D brain representation of a single subject in stereotaxic space showing: (1) Heat maps representing overlap statistics of the reported coordinates from published fMRI (14) and TMS (7) studies based on the methods we describe in our Supplementary Materials S5 File; (2) Peak activation clusters from Fitzpatrick et al. (2019) presented per hemisphere (in green). Activation above a minimum $t(22) = 4.01$ is shown. (Lower panel) The same information is also shown on five axial anatomical MRI slices of this same individual. LH/L: Left hemisphere. RH: Right hemisphere.

methods. The method we used, cTBS, is thought to induce an inhibitive stimulation aftereffect; a decrease in the excitatory potential of the neurons underneath where the stimulation was applied [37]. The effects are expected to show a particular time course, gradually increasing in intensity and peaking sometime (approximately five minutes) after stimulation; in our study, at the time where participants began performing the task. This differs from the single-pulse TMS methods used by Oliveira and colleagues [19], applied during task performance and

expected to immediately disrupt the processing of underlying neural events. Perhaps this can explain our inconsistent results; it is possible that the acute disruption of processing within the PPC during premovement planning, on a trial-by-trial basis, is necessary to cause a change in hand choice. In other words, reducing the excitatory potential of the neurons within the PPC, as we have presumed to have done in the current study, may be insufficient to drive a change in hand choice. This interpretation compels us to reject the PPIC model of hand choice, and the competitive push-pull dynamics between the two hemispheres of the PPC that we have proposed to underpin this process.

Recent results in non-human primates add to our understanding of how single-pulse TMS compared to cTBS alters neural processing within posterior parietal neurons. Targeting parietal area PFG, Romero et al. [38] examined the aftereffects of cTBS on single-neuron responses (see also [39], focused on behavioural effects). Although some neurons showed a complex pattern of both hypo- and hyperexcitability changes, population-level results showed the expected pattern of reduced excitability following cTBS. These effects were characterised by a temporal profile that was delayed in its onset and that gradually increased in magnitude, reaching peak suppression at about 30–40 minutes post cTBS. This corresponds well with the direction and timing of the aftereffects described in human participants following cTBS to primary motor cortex ([20]; for review see [37]).

In a separate study, this same group examined the neural effects of single-pulse TMS, again targeting area PFG [40]. Single-pulse TMS was applied while performing a reach-to-grasp task. TMS induced an early pronounced burst of increased neural firing within task-related neurons, followed by a longer phase of suppressed activity (although still less than one second), and these effects were accompanied by increased movement times required for grasping. The results are consistent with the idea that single-pulse TMS causes an immediate and short-lasting disruption of task-related neural processing, as opposed to the significantly delayed, slow-rising and relatively sustained changes in neural excitability that follow cTBS.

The argument that changes in PPC excitability are insufficient to drive changes in hand choice, however, conflicts with the findings of Hirayama et al. [22]. Using a behavioural protocol similar to that of the current study, Hirayama and colleagues [22] measured hand choice behaviour before, during, and after tDCS was applied bilaterally to the PPC. The findings reveal an effect of tDCS that, as expected, was specific to the post-stimulation phase. Participants were more likely to use their left hand when cathodal stimulation, which is thought to generate aftereffects that lead to decreased excitability, was applied to the left hemisphere PPC while anodal stimulation, which is thought to generate aftereffects that lead to increased excitability, was applied the right hemisphere PPC. The reverse electrode configuration, with the cathode over the right PPC and the anode over the left PPC, had no effect.

Like cTBS, tDCS changes the excitability state of the cortex for a period of time following stimulation. Moreover, both cTBS and tDCS exert aftereffects that appear to reflect common mechanisms (for review of tDCS, see [41]; for review of theta burst TMS, see [37]). Changes in synaptic plasticity as a consequence of either cTBS or tDCS depend on NMDA receptor mechanisms and Ca2+ signalling [42–45]. The findings of Hirayama et al. [22] suggest that changing the excitability of the PPC can produce changes in hand choice. It is therefore surprising that we find no evidence for a change in hand choice following cTBS to unilateral PPC.

The reason why Hirayama et al. [22] found effects that were asymmetrical remains unclear. Their preferred interpretation was that the asymmetry of their results was due to the fact that their participants were all right-handed, and as a consequence, there was little 'room' for tDCS to push hand choice in a direction that led to increased use of the right hand. A similar possibility was raised by Oliveira et al. [19] to explain why they also only found a change in hand choice in the direction of increased left hand use. As an alternative account, however,

Hirayama et al. [22] noted that perhaps the asymmetry of their effects reflects the action of one type of stimulation, cathodal or anodal (and not both), working over one hemisphere. This would suggest that the mechanisms underpinning hand selection are lateralised within the PPC, and that only one type of stimulation has an influence on them. For example, perhaps the cathodal-to-the-left-PPC arrangement was the cause of their tDCS effects, showing increased left-hand use. If only cathodal (and not anodal) stimulation drives a change in hand choice, and only when applied to the left (and not the right) PPC, this would explain why the reverse electrode configuration had no effect. Indeed, this would align with the results of Oliveira et al. [19] suggesting a left-lateralised role for the PPC in hand choice, and, more broadly, with the evidence from other studies suggesting that the premotor areas responsible for action selection are predominantly left-lateralised [46–49]. The problem is, however, the asymmetrical effects observed by Hirayama et al. [22] could have instead been driven by the anodal-to-the-right-PPC arrangement, or, as the authors argued, a sensitivity confound related to the inclusion of only right-handers (as noted above). Future studies involving both left- and right-handers, comparing separate unilateral and bilateral stimulation conditions are necessary to tease apart these potential explanations.

A difficult challenge with the use of cTBS is that unless applied to primary motor cortex confirmation of the expected aftereffects is difficult to achieve. Expected direction (excitatory or inhibitory) and timing of cTBS aftereffects are based on resulting changes in measures of corticospinal excitability following stimulation of the primary motor cortex ([20]; for review, see [37]). These measurements are based on well-established combined single-pulse TMS and electromyographic recordings. Characterising the aftereffects of cTBS to brain areas outside of the primary motor cortices is much more difficult (although primary visual areas are a notable exception), and interpretation is often not straightforward. This may come in the way of drawing inferences on the basis of behaviour; for which, we would like to note, there are examples involving cTBS to the PPC wherein the behavioural effects are consistent with reduced excitability [50–52]. Otherwise, cTBS may be combined with neuroimaging methods; examples include the use of EEG [53, 54], fMRI [55, 56], and MRS [57]. We did not pair our current study with these additional methods, and thus are unable to evidence whether the application of cTBS to the left/right PPC had the expected reduced-excitability aftereffects.

Of additional concern, the direction of the aftereffects of cTBS to primary motor cortex have also been found to vary considerably between individuals, with some individuals even showing the reverse effects–increased excitability. The cause of this variability is unknown; many different factors have now been implicated (for review see [37]; and see [58] for important methodological considerations), including the particular structural arrangement of the cell types within primary motor cortex [59, 60]. If the same kind of variability seen in research involving cTBS to primary motor cortex exists for other brain areas, this is a concern. If our group of participants happened to comprise a mixture of inhibitory (expected) and excitatory (unexpected) 'responders', then the effects of cTBS at the group-level may have been obscured. Clearly, future work will benefit from a better understanding of the potential interparticipant water in the direction of aftereffects of cTBS when applied to brain areas outside of the primary motor cortex.

Unfortunately, even if the particular direction of aftereffects after cTBS to primary cortex were known for a given individual, it is yet unknown whether and how this relates to the direction of aftereffects on other brain areas within that same individual. Nonetheless, perhaps it would be of value for future studies to characterise the direction of cTBS aftereffects on primary motor cortex and use this information to stratify participants for analyses of cTBS effects after its application to other brain areas.

## 4.2 Differences in brain area targeting methods

An inherent challenge when comparing across brain stimulation studies is that it is difficult to be sure that the same brain areas were stimulated. Oliveira et al. [19] used individual-level anatomical MRI data to target both the left and right PPC. They report centring the TMS coil over the posterior part of the intraparietal sulcus, just anterior to the parieto-occipital sulcus. The handle orientation of the coil was aligned with the posterior-anterior axis, directed posteriorly. They used a 70mm figure-of-eight coil, as we did in the current study. Hirayama et al. [22] centred their stimulation electrodes (sized 5 cm x 7 cm) over P3 and P4 according the International 10–20 system (and also provide a simulation of cortical current flow based on their stimulation parameters and electrode placements).

In our study, like Oliveira et al. [19], we also used individual-level anatomical MRI data to position the TMS coil; yet, this was first guided by functional results. Specifically, per individual and for each hemisphere, we first positioned the coil according to the standard-space coordinates of our previous group-level fMRI results showing evidence for the preferential involvement of the posterior intraparietal and superior parietal cortex in hand choice [16]. Second, and only if necessary, we then adjusted the coil according to the individual's anatomy so that the trajectory of stimulation would pass through the medial bank of the intraparietal sulcus, within the superior parietal cortex.

We consider our targeting approach as comparable to that of Oliveira et al. [19]. We both intended to target an area involved with the planning and control of arm movements, sometimes referred to as the "parietal reach region" (for reviews, see [61, 62]). By also using our prior functional data related to hand choice (during a reach task) we hoped to further increase the specificity of our localisation approach, to more precisely target the part of the PPC that was most likely to be involved in hand choice according to these prior findings. Nevertheless, it is possible that the differences in our targeting methods may have resulted in the stimulation of separate functional parts of the PPC, and this may have contributed to our discrepant results.

To possibly assist with further research in this area, we provide a synthesis of functional-localisation data from various previous fMRI and TMS studies involving reaching, shown alongside our recent fMRI results [16] that were used as a guide for TMS coil localisation in the current study (Fig 5; see S5 File Comparison of reaching studies). We hope that this visualisation provides a resource that will be of value for future brain stimulation studies looking to target reach-related PPC.

## 4.3 Differences in behavioural protocols

As noted above, the behavioural protocol we used in the current study is modelled after that of Oliveira et al. [19], and is also very similar to that of Hirayama et al. [22]. All three investigations had participants reach to visible targets presented on either side of hemispace, and participants could themselves choose which hand to use on a given trial. All three studies also used the same two additional 'catch' trials, involving either reaching to two-targets with both hands, or reaching to fixation, again using both hands. Most importantly, our results reveal what we consider as a kind of behavioural 'signature' of this paradigm, a marker of the deliberation process underlying hand choice–namely, that response times to initiate actions are significantly greater when reaching to targets near the PSE relative to when reaching to targets that are further out laterally, to one side of space. These results are consistent with expectations related to intermanual differences in biomechanical and energetic costs according to where movements are directed in space—for more lateralised targets, using the hand on the same side of space is associated with lower costs [63–66]—, and that the deliberation process underlying action

choices involves resolving competitive influences that represent these and other factors [4, 18, 25, 67, 68].

These same effects were reported by Oliveira et al. [19], and others [69–71]; and, are robust to seemingly minor differences in protocols—for example, we see these effects in the current study using a vertically oriented display for target presentation, whereas in Oliveira et al. [19], and in our own prior work documenting similar effects [25], targets were presented in the horizontal plane. We consider these effects as key support for the view that our current protocol was comparable to that of Oliveira et al. [19]; and, as such, that participants in either study were likely to have relied on similar brain mechanisms to decide which hand to use to perform the task from trial to trial. Hirayama et al. [22], unfortunately, do not report tests of response time differences according to target position.

Regardless of our own position on this, it is worth noting the differences between our behavioural protocol and those of Oliveira et al. [19] and Hirayama et al. [22]. Hirayama et al. [22] had participants respond within 650ms of target onsets, whereas in both the current study and Oliveira et al. [19] participants were instructed to respond as quickly and accurately as possible, without a time restriction implemented. Oliveira et al. [19] also gamified their reaching protocol in that participants were awarded points according to speed and accuracy, and points were taken away for catch trials that were performed incorrectly. Hirayama et al. [22], instead, provided feedback about the endpoint accuracy of each reaching movement, on a trial-by-trial basis; different tones were played for successful and unsuccessful reaches, defined as whether the reach ended within the radius of the target, tracked using a motion capture system with markers attached to the index finger of each hand. We, conversely, did not provide performance feedback to participants. This may well have had a meaningful influence on our results, as, perhaps, participants in our study were relatively less motivated.

Another difference was that in both Oliveira et al. [19] and Hirayama et al. [22] participants could not see their limbs during reaching, only a visual representation of their hands, whereas vision was fully available in our study. Whether this difference was an important factor remains unclear. With respect to the *control* of reaching, the role of the posterior parietal cortex is not limited to when visual feedback of the moving arm and hand is unavailable ([72, 73]; for review see [61, 74]). Nonetheless, it is difficult to know whether having full vision of the limbs in our study diminished the effects of cTBS; perhaps the brain is better able to compensate under these conditions. This possibility requires direct testing.

Finally, participants in our study also completed considerably more trials relative to either Oliveira et al. [19] or Hirayama et al. [22]. It is possible that the task became 'overlearned' in our study, and that this made our participant's hand choice behaviour more resistant to influence from brain stimulation.

The current study used a larger sample size than either Oliveira et al. [19] and Hirayama et al. [22], who tested 10 and 16 participants, respectively. All three studies tested right-handers.

## 4.4 Concluding remarks

We recognise that our current results are challenging to interpret, yet are nonetheless confident that our report will stand as a useful record and point of comparison for future investigations. In light of the methodological considerations discussed above, our results do not directly refute prior evidence; however, in our view, the question of whether non-invasive brain stimulation to the posterior parietal cortex is capable of modulating choices about which hand to use to perform actions remains unresolved, and is of clear importance to further pursue. Developing an offline brain stimulation protocol that can be used to boost the likelihood that a

patient will use a particular limb to perform rehabilitation exercises, for example, could offer a novel way to improve their functional outcomes.

## Supporting information

**S1 File. Supplementary statistical analyses.** Hand choice.
(DOCX)

**S2 File. Supplementary statistical analyses.** Response times.
(DOCX)

**S3 File. Participant errors.**
(DOCX)

**S4 File. Additional analyses.** No-cTBS baseline.
(DOCX)

**S5 File. Comparison of reaching studies.**
(DOCX)

**S6 File. Post-stimulation questionnaire data.**
(DOCX)

## Acknowledgments

We thank Dave McKiernan and James Morgan for their technical support.

## Author Contributions

**Conceptualization:** Aoife M. Fitzpatrick, Kenneth F. Valyear.

**Data curation:** Aoife M. Fitzpatrick, Kenneth F. Valyear.

**Formal analysis:** Aoife M. Fitzpatrick, Neil M. Dundon, Kenneth F. Valyear.

**Investigation:** Aoife M. Fitzpatrick, Kenneth F. Valyear.

**Methodology:** Aoife M. Fitzpatrick, Kenneth F. Valyear.

**Project administration:** Aoife M. Fitzpatrick, Kenneth F. Valyear.

**Software:** Neil M. Dundon.

**Supervision:** Kenneth F. Valyear.

**Visualization:** Aoife M. Fitzpatrick, Kenneth F. Valyear.

**Writing – original draft:** Aoife M. Fitzpatrick, Kenneth F. Valyear.

**Writing – review & editing:** Aoife M. Fitzpatrick, Neil M. Dundon, Kenneth F. Valyear.

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
