## [Decision Letter · Decision Letter 0]

20 Jun 2022

PONE-D-22-13883No effects of offline high frequency transcranial magnetic stimulation to posterior parietal cortex on the choice of which hand to use to perform a reaching task.PLOS ONE

Dear Dr. Fitzpatrick,

Thank you for submitting your manuscript to PLOS ONE. After careful consideration, we feel that it has merit but does not fully meet PLOS ONE’s publication criteria as it currently stands. Therefore, we invite you to submit a revised version of the manuscript that addresses the points raised during the review process.

We look forward to receiving your revised manuscript.

Kind regards,

Victor Frak, MD, Ph.D

Academic Editor

PLOS ONE

Journal Requirements:

3. Please change "female” or "male" to "woman” or "man" as appropriate, when used as a noun (see for instance https://apastyle.apa.org/style-grammar-guidelines/bias-free-language/gender).

5. Please note that in order to use the direct billing option the corresponding author must be affiliated with the chosen institute. Please either amend your manuscript to change the affiliation or corresponding author, or email us at plosone@plos.org with a request to remove this option.

Reviewers' comments:

Reviewer's Responses to Questions

**Comments to the Author**

1. Is the manuscript technically sound, and do the data support the conclusions?

Reviewer #1: Partly

Reviewer #2: Yes

2. Has the statistical analysis been performed appropriately and rigorously? 

Reviewer #1: Yes

Reviewer #2: Yes

3. Have the authors made all data underlying the findings in their manuscript fully available?

Reviewer #1: Yes

Reviewer #2: Yes

4. Is the manuscript presented in an intelligible fashion and written in standard English?

Reviewer #1: Yes

Reviewer #2: Yes

5. Review Comments to the Author

Reviewer #1: Thanks for the manuscript. The major concern that I have is with the potency of cTBS with 600 pulses at the posterior parietal cortex (Huang et al. (2005) is on human motor cortex). Also, other studies (ref. 31, 32) referenced on the TMS of the posterior parietal cortex are not cTBS. In fact, cTBS effects are known to be inconsistent with high inter-subject variability (https://pubmed.ncbi.nlm.nih.gov/32758665/). Therefore, first step should be a dose-response validation of the cTBS intervention with neurophysiological testing before using the method for probing a scientific question. Also, I don't agree with the title, "No effects of offline high frequency transcranial magnetic stimulation to posterior parietal cortex on the choice of which hand to use to perform a reaching task," since high frequency TMS can also mean other rTMS paradigms. Then, the Discussion section somewhat addressed these concerns with cTBS; however, in my opinion, the scientific statement countering published results is unwarranted.

The manuscript can be written differently for example to highlight the failure of cTBS method with 600 pulses at the posterior parietal cortex on modulating the hand choice. Moreover, the orientation of the coil may be relevant where the neural mechanisms underlying the effects of cTBS are poorly understood (network mechanisms may be relevant: https://pubmed.ncbi.nlm.nih.gov/23941616/).

Reviewer #2: PONE-D-22-13883

Review

General comments

The manuscript is very well written, with commendable fluency, straightness and assertiveness. Although the results were not those predicted by the authors, they were clear in their description, providing the tools for the reader to understand the study and the possible reasons for the results obtained. The study, therefore, is of great relevance for understanding the differences between current forms of non-invasive brain stimulation, namely, transcranial magnetic stimulation (TMS), high-frequency repetitive continuous theta burst stimulation (cTBS) and transcranial direct current stimulation (tDCS). It also contributes to the discussion about the parameters of measurement and application of cTBS, particularly when applied to regions not directly related to the primary motor cortex.

I only have few suggestions for the manuscript. I present them below, separated by section of the manuscript.

Introduction

I enjoyed reading the introduction, although I missed a greater number of references - or a brief discussion about the scarcity of productions related to the topic. The development of the introduction is well done, fluid and dynamic. The presentation of the model idealized by the authors is well-organized and sufficiently detailed.

In lines 58-62, you cite two models based on competition between two neuronal populations. The excerpt that extends from line 62 to line 69, however, makes several important claims, but there are no references to them. If this excerpt is an explanation of the models cited in lines 58-62, it is necessary to mention the link between the two excerpts.

Methods

The inclusion of the work hypotheses on the aspredicted.org portal is an interesting work strategy that should be endorsed, as well the sample sizing through a free and accessible tool.

The use of the Waterloo Handedness Inventory is interesting. According to the article cited, "the type of questionnaire used in the present investigation allows subjects to indicate both the amount or degree of their hand preference and the direction of their hand preference" (1). Did you rate the degree of hand preference (consistency) of the participants? Although Steenhuis et colleagues (1) comments that the population's manuality consistency is usually high, I would like to know if your sample was homogeneous in terms of consistency and if the left-handers in the study had a high or low manuality consistency.

The description of the stimulation protocol is well done and detailed. As the two articles used to justify the positioning of the coil-handle are from the same research group and quite old – 2008 and 2010 – I believe that only the second reference is enough, or I suggest that the second is kept and a more recent reference is added.

Regarding the behavioral test, Oliveira et al. (2) state that "the instructions emphasized that the responses should be initiated and completed as fast as possible in a single smooth movement, and that end-point errors need not be corrected". Was there a similar instruction in your study?

Completing my ‘Methods’ commentaries, there is a significant change in the design of the work compared to the work developed by Oliveira (2) and Valyear (3) and collaborators, which now uses a touchscreen monitor and different angles of stimulus presentation, although the difference is minimal. I believe you could briefly explain why you chose to change the angles in relation to the studies cited.

Results

The results are well described and organized, and their presentation is clear and easily understandable. I only have two comments about the Figures 4 and 5.

Figure 4 does not clearly illustrate the difference between the RTs of targets close to the PSE and in extreme positions, although it provides more information about the data distribution. I think the representation of the difference between the RTs in the two conditions benefits more from another graphical representation strategy. The inset does not adequately illustrate the collapsed difference between the RTs, and does not provide units that allow its dimensioning..

Figure 5 is very good, but the green line that connects the highlighted region to the highlight frame in the posterior and supero-posterior views interferes with the visualization. I suggest also delimiting the highlighted region and connect it to the highlight frame with solid lines at the vertices, demonstrating the applied zoom.

Discussion

The discussion follows the same pattern of organization and quality of the manuscript. The authors are thorough in analyzing the possible causes of the differences found between their study and similar studies that preceded it. They assess the impact that the form of stimulation (cTBS, tDCS, sp-TMS) may have caused, probably constituting the main responsible for the difference between the results obtained, while considering the possible effects caused by the sample and the study design.

The influence of the vision in the previous studies and in the current study also seems to be relevant and, if it was not an important factor allied to the the difference in results, it was a factor of divergence between them, which makes the similarities and differences between their results less comparable. Thus, it would be interesting to contemplate a little more in the discussion the bias that the vision may have brought to the study results.

References

1. Steenhuis, R. E., & Bryden, M. P. (1989). Different dimensions of hand preference that relate to skilled and unskilled activities. Cortex; a journal devoted to the study of the nervous system and behavior, 25(2), 289–304. https://doi.org/10.1016/s0010-9452(89)80044-9

2. Oliveira, F. T., Diedrichsen, J., Verstynen, T., Duque, J., & Ivry, R. B. (2010). Transcranial magnetic stimulation of posterior parietal cortex affects decisions of hand choice. Proceedings of the National Academy of Sciences of the United States of America, 107(41), 17751–17756. https://doi.org/10.1073/pnas.1006223107

3. Valyear, K. F., Fitzpatrick, A. M., & Dundon, N. M. (2019). Now and then: Hand choice is influenced by recent action history. Psychonomic bulletin & review, 26(1), 305–314. https://doi.org/10.3758/s13423-018-1510-1

6. PLOS authors have the option to publish the peer review history of their article (what does this mean?). If published, this will include your full peer review and any attached files.

Reviewer #1: No

Reviewer #2: **Yes: **Ronaldo Luis da Silva

---

## [Author Response · Author response to Decision Letter 0]

1 Aug 2022

*Please see attached for formatted responses: Fitzpatricketal_PLOSONE_R01_Responses.docx

Responses to Reviewers

Originally submitted manuscript number: PONE-D-22-13883

We appreciate the reviewers’ feedback. We have made changes to the manuscript to address and incorporate their comments and suggestions. We believe these changes have improved the manuscript, and would like to thank the reviewers for their help. 

Reviewers’ comments are in normal text and our point-by-point responses are in blue coloured text. 

Reviewers' comments:

Reviewer 1: 

Thanks for the manuscript. The major concern that I have is with the potency of cTBS with 600 pulses at the posterior parietal cortex (Huang et al. (2005) is on human motor cortex). Also, other studies (ref. 31, 32) referenced on the TMS of the posterior parietal cortex are not cTBS. In fact, cTBS effects are known to be inconsistent with high inter-subject variability (https://pubmed.ncbi.nlm.nih.gov/32758665/). Therefore, first step should be a dose-response validation of the cTBS intervention with neurophysiological testing before using the method for probing a scientific question. 

Also, I don't agree with the title, "No effects of offline high frequency transcranial magnetic stimulation to posterior parietal cortex on the choice of which hand to use to perform a reaching task," since high frequency TMS can also mean other rTMS paradigms. 

Then, the Discussion section somewhat addressed these concerns with cTBS; however, in my opinion, the scientific statement countering published results is unwarranted.

The manuscript can be written differently for example to highlight the failure of cTBS method with 600 pulses at the posterior parietal cortex on modulating the hand choice. 

Moreover, the orientation of the coil may be relevant where the neural mechanisms underlying the effects of cTBS are poorly understood (network mechanisms may be relevant: https://pubmed.ncbi.nlm.nih.gov/23941616/).

(R1-1) We thank the reviewer for their input. We recognize two concerns — (1) the efficacy of cTBS; (2) differences in TMS protocols across relevant studies (i.e., Oliveira et al. 2010; Hirayama et al. 2021) — and, we fully agree, these are important concerns. 

Efficacy of cTBS

While we agree that obtaining a dose-response profile describing the neurophysiological effects cTBS when applied to the posterior parietal cortex would be extremely valuable, to do so presents various challenges that far exceed the scope and capacity of the current study. Perhaps most challenging, basic knowledge is lacking regarding both the type of neurophysiological measurements that would be most useful and how the resultant measurements should be interpreted. 

For example, in our experiment, perhaps one could use fMRI immediately following cTBS to try and better understand its effects on the posterior parietal cortex. However, would the BOLD signal be suppressed or increased? Perhaps, as a consequence of reduced excitatory potential, a cTBS-targeted brain area is forced to ‘work harder’, thereby leading to greater metabolic demands and increased BOLD signal. Yet, these predictions are unclear (to us), and require extensive further development. 

In short, aside from the fact that additional experiments would be required to meet this request, the principles necessary to guide the design of those experiments, and their results interpretation, have not been developed. This is beyond the scope of the current study. 

We would also like to note that we provide a discussion of these challenges, see p. 26-27.

Related, the reviewer expresses concerns regarding the high degree of variability across individuals in the effects of cTBS. We agree completely, this is an important concern. Yet, we would like to emphasise that we address this concern directly, and, we feel, are clear and transparent about both the importance of recognising this challenge, and its potential impact on our findings. We were unaware, however, of the valuable reference provided by the reviewer, and now include this reference in our discussion, accordingly (see p.27, ref 58). 

For convenience, we have copied this part of our discussion below:

Of additional concern, the direction of the aftereffects of cTBS to primary motor cortex have also been found to vary considerably between individuals, with some individuals even showing the reverse effects – increased excitability. The cause of this variability is unknown; many different factors have now been implicated (for review see 38; and see 58 for important methodological considerations), including the particular structural arrangement of the cell types within primary motor cortex (59,60). If the same kind of variability seen in research involving cTBS to primary motor cortex exists for other brain areas, this is a concern. If our group of participants happened to comprise a mixture of inhibitory (expected) and excitatory (unexpected) ‘responders’, then the effects of cTBS at the group-level may have been obscured. Clearly, future work will benefit from a better understanding of the potential interparticipant water in the direction of aftereffects of cTBS when applied to brain areas outside of the primary motor cortex. 

Unfortunately, even if the particular direction of aftereffects after cTBS to primary cortex were known for a given individual, it is yet unknown whether and how this relates to the direction of aftereffects on other brain areas within that same individual. Nonetheless, perhaps it would be of value for future studies to characterise the direction of cTBS aftereffects on primary motor cortex and use this information to stratify participants for analyses of cTBS effects after its application to other brain areas. 

As a final concern about the effects of cTBS, the reviewer draws attention to uncertainty regarding the importance of TMS coil orientation. While we agree that this indeed could be important, we again point to the challenges associated with assessing the effects of cTBS outside of the primary motor cortex, and the general lack of knowledge regarding which combination of methods may be best to do so, as discussed above. Certainly, as this field develops and these fundamental problems are better addressed, systematic investigation of the potential importance of coil orientation would be of value. 

Differences in TMS protocols across relevant studies (i.e., Oliveira et al. 2010; Hirayama et al. 2021).

The reviewer raises concerns about the fact that those prior studies which provide causal evidence for the role of the posterior parietal cortex in deciding which hand to use to perform actions used different brain stimulation methods than cTBS. 

We absolutely agree, this point is essential to recognise and appreciate. Indeed, our Discussion is predominately devoted to addressing this point. We provide a detailed, thorough and transparent, review of the methodological differences between these studies and our own, emphasising the challenges associated with interpreting results derived from single-pulse TMS, tDCS, and cTBS (as used in our study). This is reflected in our discussion subheadings. 

Critically, in no way did we intend to suggest that our current results directly refute this prior work. Instead, it was our intention to draw attention to what we feel are the key factors to consider, with the view that our report will serve as a useful guide for future investigations in this area. Thanks to this reviewer’s feedback, we have now made this view more explicit in our concluding remarks (see p. 31-32).

Additionally, we have made a significant adjustment to the outset of our Discussion; see p. 22. This will orient the reader to this critical point right away, so that its importance in the preceding discussion is not likely to be missed. 

Finally, the reviewer suggests that we change the manuscript’s title to, in particular, clarify the type of high frequency stimulation used. We have made this change accordingly (see revised manuscript).

We would, however, like to express some reservation regarding this change. Our worry is that by including too much information about the methods, the title becomes difficult to understand. We do clearly specify the type of TMS protocol we use in the first line of our Abstract. 

We are happy for the Editor to exert their discretion on this point. 

Reviewer #2: 

General comments

The manuscript is very well written, with commendable fluency, straightness and assertiveness. Although the results were not those predicted by the authors, they were clear in their description, providing the tools for the reader to understand the study and the possible reasons for the results obtained. The study, therefore, is of great relevance for understanding the differences between current forms of non-invasive brain stimulation, namely, transcranial magnetic stimulation (TMS), high-frequency repetitive continuous theta burst stimulation (cTBS) and transcranial direct current stimulation (tDCS). It also contributes to the discussion about the parameters of measurement and application of cTBS, particularly when applied to regions not directly related to the primary motor cortex.

I only have few suggestions for the manuscript. I present them below, separated by section of the manuscript.

Introduction

I enjoyed reading the introduction, although I missed a greater number of references - or a brief discussion about the scarcity of productions related to the topic. The development of the introduction is well done, fluid and dynamic. The presentation of the model idealized by the authors is well-organized and sufficiently detailed. In lines 58-62, you cite two models based on competition between two neuronal populations. The excerpt that extends from line 62 to line 69, however, makes several important claims, but there are no references to them. If this excerpt is an explanation of the models cited in lines 58-62, it is necessary to mention the link between the two excerpts.

(R2-1) Thank you for this comment, and in general, for your thoughtful and constructive feedback. We have revised this section, making the link explicit between these detailed claims and the two referenced models from which they are described. See line 62 of our revised manuscript. 

Methods

The inclusion of the work hypotheses on the aspredicted.org portal is an interesting work strategy that should be endorsed, as well the sample sizing through a free and accessible tool. The use of the Waterloo Handedness Inventory is interesting. According to the article cited, "the type of questionnaire used in the present investigation allows subjects to indicate both the amount or degree of their hand preference and the direction of their hand preference" (1). Did you rate the degree of hand preference (consistency) of the participants? Although Steenhuis et colleagues (1) comments that the population's manuality consistency is usually high, I would like to know if your sample was homogeneous in terms of consistency and if the left-handers in the study had a high or low manuality consistency.

(R2-2) Thanks for the reviewer’s comments, here. We have updated the manuscript to include this information (see p. 9, of the manuscript).

Our group of right-handers shows considerable interparticipant variation in their scores. Below, we plot these data, for the purpose of this response letter.

Figure A, R2-2: Boxplot of Waterloo Handedness Questionnaire scores from our group of right-handed participants. 

In addition, we provide this information for our left-handed participants, reported on p.9 in the manuscript. 

The description of the stimulation protocol is well done and detailed. As the two articles used to justify the positioning of the coil-handle are from the same research group and quite old – 2008 and 2010 – I believe that only the second reference is enough, or I suggest that the second is kept and a more recent reference is added.

(R2-3) We have made the recommended changes, updating our manuscript to include a recent reference (Breveglieri et al., 2021) which also uses this TMS coil orientation to disrupt the processing of posterior parietal cortex in a reaching task (see p. 12). We prefer to keep our initial two references as well, however, as, although they are from the same group, they demonstrate distinct and relevant effects at this same coil orientation. 

Regarding the behavioral test, Oliveira et al. (2) state that "the instructions emphasized that the responses should be initiated and completed as fast as possible in a single smooth movement, and that end-point errors need not be corrected". Was there a similar instruction in your study?

(R2-4) Participants were instructed to reach to the target as quickly and as accurately as possible, with no explicit instruction regarding the possible correction of their reach trajectories. We have now updated the manuscript to make this explicit (see line 272-73, Methods).

Completing my ‘Methods’ commentaries, there is a significant change in the design of the work compared to the work developed by Oliveira (2) and Valyear (3) and collaborators, which now uses a touchscreen monitor and different angles of stimulus presentation, although the difference is minimal. I believe you could briefly explain why you chose to change the angles in relation to the studies cited.

(R2-5) The change to a touchscreen-monitor and vertical orientation of targets was made with the intention of using the touchscreen to measure movement times and end-point errors. We had hoped that by using an affordable and easy-to-use technology, like a touchscreen, our work could better translate to clinical applications (e.g., stroke rehab). Yet, these measurements turned out to be unreliable. There were too many missing datapoints; and, although piloting indicated that this could be mitigated to some extent by instructing participants to end their movements by pressing and holding their finger flat to the screen on contact, we were concerned that the resulting actions were unnatural. And, even with this explicit instruction, endpoint measurements would still sometimes be missed. We opted to revert to the more natural reaching instructions. 

The change in target angles was done to equate the distances between targets and both hands; this was motivated by comments from colleagues, as a control for different reach distances. Critically, our same pilot behavioural experiments mentioned above showed the expected pattern of increased response times to initiate actions for targets near the middle (and PSE) relative to the edges of the display—a result that we consider as an important validation of the paradigm, as we discuss in the manuscript (p. 29-30, Discussion). Notably, too, our fMRI experiment (Fitzpatrick et al., 2019) used a vertically-oriented target display. 

Results

The results are well described and organized, and their presentation is clear and easily understandable. I only have two comments about the Figures 4 and 5. Figure 4 does not clearly illustrate the difference between the RTs of targets close to the PSE and in extreme positions, although it provides more information about the data distribution. I think the representation of the difference between the RTs in the two conditions benefits more from another graphical representation strategy. The inset does not adequately illustrate the collapsed difference between the RTs, and does not provide units that allow its dimensioning.

Figure 5 is very good, but the green line that connects the highlighted region to the highlight frame in the posterior and supero-posterior views interferes with the visualization. I suggest also delimiting the highlighted region and connect it to the highlight frame with solid lines at the vertices, demonstrating the applied zoom. 

(R2-6) These are great suggestions that we think helped to improve the clarity and presentation of these Figures. Thank you. We have made the suggested changes. Please see revised Figures 4 and 5. 

Discussion

The discussion follows the same pattern of organization and quality of the manuscript. The authors are thorough in analyzing the possible causes of the differences found between their study and similar studies that preceded it. They assess the impact that the form of stimulation (cTBS, tDCS, sp-TMS) may have caused, probably constituting the main responsible for the difference between the results obtained, while considering the possible effects caused by the sample and the study design. The influence of the vision in the previous studies and in the current study also seems to be relevant and, if it was not an important factor allied to the difference in results, it was a factor of divergence between them, which makes the similarities and differences between their results less comparable. Thus, it would be interesting to contemplate a little more in the discussion the bias that the vision may have brought to the study results.

(R2-7) We are thankful for the reviewer’s comments on this, and for their positive feedback regarding our Discussion, generally. We have added some discussion of this point, to draw better attention to the possibility that this difference may have been an important factor. For convenience, we copy our changes below (taken from p. 31 of the revised manuscript):

Another difference was that in both Oliveira et al. (19) and Hirayama et al. (22) participants could not see their limbs during reaching, only a visual representation of their hands, whereas vision was fully available in our study. Whether this difference was an important factor remains unclear. With respect to the control of reaching, the role of the posterior parietal cortex is not limited to when visual feedback of the moving arm and hand is unavailable (71,72; for review see 60,73). Nonetheless, it is difficult to know whether having full vision of the limbs in our study diminished the effects of cTBS; perhaps the brain is better able to compensate under these conditions. This possibility requires direct testing. 

References

Breveglieri, R., Bosco, A., Borgomaneri, S., Tessari, A., Galletti, C., Avenanti, A., & Fattori, P. (2021). Transcranial magnetic stimulation over the human medial posterior parietal cortex disrupts depth encoding during reach planning. Cerebral Cortex, 31(1), 267-280.

---

## [Decision Letter · Decision Letter 1]

13 Sep 2022

Hand choice is unaffected by high frequency continuous theta burst transcranial magnetic stimulation to the posterior parietal cortex.

PONE-D-22-13883R1

Dear Dr. Valyear,

We’re pleased to inform you that your manuscript has been judged scientifically suitable for publication and will be formally accepted for publication once it meets all outstanding technical requirements.

Kind regards,

Victor Frak, MD, Ph.D

Academic Editor

PLOS ONE

Additional Editor Comments (optional):

Dear Ken,

The reviewers confirm that the necessary changes were made in the second version of the manuscript and your production meets the requirements to be published in PLOSONE.

Thank you for submitting your very interesting work to PLOSONE.

Cordially,

Victor Frak

Reviewers' comments:

Reviewer's Responses to Questions

**Comments to the Author**

1. If the authors have adequately addressed your comments raised in a previous round of review and you feel that this manuscript is now acceptable for publication, you may indicate that here to bypass the “Comments to the Author” section, enter your conflict of interest statement in the “Confidential to Editor” section, and submit your "Accept" recommendation.

Reviewer #1: All comments have been addressed

Reviewer #2: All comments have been addressed

2. Is the manuscript technically sound, and do the data support the conclusions?

Reviewer #1: Yes

Reviewer #2: Yes

3. Has the statistical analysis been performed appropriately and rigorously? 

Reviewer #1: Yes

Reviewer #2: Yes

4. Have the authors made all data underlying the findings in their manuscript fully available?

Reviewer #1: No

Reviewer #2: Yes

5. Is the manuscript presented in an intelligible fashion and written in standard English?

Reviewer #1: Yes

Reviewer #2: Yes

6. Review Comments to the Author

Reviewer #1: Thanks. I am good with the revisions and the new title. I also agree with this major concern that needs future studies >> "Of additional concern, the direction of the aftereffects of cTBS to primary motor cortex

have also been found to vary considerably between individuals, with some individuals

even showing the reverse effects – increased excitability. The cause of this variability is

unknown; many different factors have now been implicated (for review see 38; and see

58 for important methodological considerations), including the particular structural

arrangement of the cell types within primary motor cortex (59,60)."

Reviewer #2: (No Response)

7. PLOS authors have the option to publish the peer review history of their article (what does this mean?). If published, this will include your full peer review and any attached files.

Reviewer #1: **Yes: **Anirban Dutta

Reviewer #2: **Yes: **Ronaldo Luis da Silva

---

## [Editor Report · Acceptance letter]

3 Oct 2022

PONE-D-22-13883R1 

Hand choice is unaffected by high frequency continuous theta burst transcranial magnetic stimulation to the posterior parietal cortex. 

Dear Dr. Valyear:

I'm pleased to inform you that your manuscript has been deemed suitable for publication in PLOS ONE. Congratulations! Your manuscript is now with our production department. 

Kind regards, 

on behalf of

Dr. Victor Frak 

Academic Editor

PLOS ONE